# Approach to Cancer Pain Management in Emergency Departments: Comparison of General and Oncology Based Settings

**DOI:** 10.3390/ph15070805

**Published:** 2022-06-28

**Authors:** Ilit Turgeman, Salvatore Campisi-Pinto, Maher Habiballah, Gil Bar-Sela

**Affiliations:** 1Cancer Center, Emek Medical Center, Afula 1834111, Israel; ilit.turgeman@gmail.com; 2Research Authority, Emek Medical Center, Afula 1834111, Israel; campisi.pinto@gmail.com; 3Division of Oncology, Rambam Health Care Center, Haifa 31096, Israel; maher12ha1@gmail.com; 4Technion Integrated Cancer Center, Faculty of Medicine, Technion—Israel Institute of Technology, Haifa 31096, Israel

**Keywords:** cancer, opioids, drugs, emergency department, pain

## Abstract

Cancer-related pain constitutes a dominant reason for admission to emergency services, and a significant patient and healthcare challenge. Evidence points to the rising prevalence of opioid misuse in this patient group. We sought to compare drug delivery in an oncology-dedicated emergency department (OED) and a general emergency department (GED) within the same hospital. As such, we obtained patient and drug-related data for OED and GED during a designated three-month period, and compared them using Fisher’s exact test, chi-square tests and the Mann-Whitney test. In total, 584 patients had 922 visits to emergency services (OED *n* = 479; GED *n* = 443), and were given 1478 drugs (OED *n* = 557; GED *n* = 921). Pain was a prominent chief complaint among visitors to the OED (17%) and GED (21%). Approximately a fifth of all drugs used were analgesics (OED—18.5%; GED—20.4%), however, in the GED, 51.6% (*n* = 97) were used for non-pain-related admissions, compared with 33.0% (*n* = 34) in OED. Opioid usage significantly differed between emergency settings. The GED administered three times as many intravenous opioids (*p* <0.001), a narrower spectrum of oral and intravenous drugs (*p* = 0.003) and no rapid-acting opioids, significantly fewer pain adjuvants (10.9% versus 18.7%, *p* < 0.001), and, finally, non-guideline-recommended drugs for pain, such as meperidine and benzodiazepines. Taken together, compared with the GED, the management of cancer-related pain in the OED was more personalized, and characterized by fewer intravenous opioids, enhanced diversity in drug type, route and method of delivery. Efforts should be directed toward reduction of disparities in the treatment of cancer pain in emergency settings.

## 1. Introduction

Patients with advanced cancer face unique disease and treatment related complications, and seek unplanned medical care more frequently than the general population [1]. A significant number of cancer patients worldwide experience pain that requires medical treatment during the course of their illness [2]. While opioids represent a drug of choice for cancer pain syndromes [3], with rapid-onset of action, lack of a ceiling effect and versatility in administration [4], emerging evidence points to the rising prevalence of opioid misuse in this patient group, especially in emergency care settings [5,6]. Restrictive regulations on opioid usage have generally exempted oncological settings, however some patients with cancer experience limited access as an unintended consequence of the ongoing opioid crisis [6]. On the other hand, opioid-associated cancer deaths have steadily increased in the last decade [7]. In the United States alone, abuse and misuse of opioids is estimated to cost billions of dollars per year [8]. Taken together, treatment of cancer related pain in emergency settings constitutes a substantial patient and healthcare challenge.

General emergency departments (GEDs) tend to be overcrowded and fast-paced, at-tending to large and heterogeneous populations, and emergency physicians often lack expertise in caring for complex cancer patients or knowledge of the rapidly evolving treatment landscape. Emergency management of patients with cancer by general physicians has been associated with delayed diagnoses, complications, and worse outcomes [9]. In recent years, various models for cancer-specific emergency services have developed globally, demonstrating both reduced costs and improved efficacy endpoints [10]. In Ram-bam Medical Center (RMC), an oncology-dedicated ED (OED) was created to provide a more intimate setting for patients with cancer in need of urgent care. During clinic hours, the OED contains 10–15 beds within the oncology daycare unit reserved for unplanned patient visits; when the OED capacity is reached, patients present directly to the hospital GED. We previously showed that treatment in the OED is significantly associated with reduced costs and hospitalizations compared with the GED [11]. In this study, we sought to compare drug utilization for a variety of admission types in both emergency settings. We hypothesized enhanced drug delivery in the oncology-based setting, promoting safe and effective urgent treatment for cancer patients.

## 2. Results

### 2.1. Patient Characteristics

Between the dates of April to June 2017 and during the 53 days the OED was open, 584 patients had 922 visits to emergency services (OED *n* = 479; GED *n* = 443). Patients were matched for age and gender, both groups exhibiting slight male predominance, and mean age was 62.8 and 63.7 in the OED and GED, respectively. Patients were more likely to reside outside of the hospital city, more so in the GED (71.3% vs. 60% *p* = 0.001), perhaps related to the limited OED opening hours by patient capacity. The discharge rate in the OED was significantly higher, at 71.6%, compared with 44.7% in the GED, however 18.8% of patients in the OED were referred for continued care in the GED (*p* < 0.001). Patients were more likely to return to the OED, with 38% returning over three times in the designated period compared with 23.5% in the GED (*p* < 0.001). In the OED, the most frequent diagnosis was gastrointestinal cancer, followed by lung, genitourinary and breast cancer, respectively. Patient characteristics are delineated in Table 1. 

### 2.2. Chief Complaints

Of ten possible chief complaints, groups were matched for all except admission for invasive procedures, such as paracentesis, which were more commonly performed in the OED, as well as oncological emergencies and neurological symptoms. We previously showed that oncological emergencies and neurological symptoms are more suited for treatment in the GED due to the multidisciplinary environment and availability of complex diagnostic and therapeutic modalities.11 Cancer pain was the most prominent chief complaint in the GED (21%) and second to invasive procedures in the OED (17%), but well balanced between the OED and GED groups (*p* = 0.117). General deterioration and respiratory and gastrointestinal complaints, were also common in both emergency settings. Fewer admissions were required to address nausea/vomiting and hemorrhagic/thromboembolic events. Chief complaints are described in Table 1.

### 2.3. Drug Utilization

#### 2.3.1. Analgesics

In total, 1478 drugs were administered in the two emergency settings (OED *n* = 557; GED *n* = 921, and their distribution is demonstrated in Table 2. Of these, 18.5% of drugs utilized in the OED and 20.4% in the GED were direct analgesics. Among the analgesics, the use of oral opioids, intravenous opioids and rapid onset opioids in OED was significantly different from GED (*p* < 0.001, *p* = 0.003, *p* = 0.003, respectively). For example, among the oral opioids, the use of oxycodone acetaminophen was more prevalent in GED 51.3% (*n* = 58) than OED 8% (*n* = 9); in the OED, enhanced diversity in drug mechanism of action, dosage, schedule (fixed or as needed), and method of delivery (syrup or pill), is evident. Similarly, among the intravenous opioids, the use of morphine was more prevalent in GED 63.1% (*n* = 41) than OED 16.9% (*n* = 11), and only physicians in the GED administered meperidine, an opioid not recommended for the treatment of oncological patients. The more modern rapid-onset opioid analgesics were only administered in the OED. On the other hand, the GED and OED showed non-dissimilar administration of non-opioid analgesics, however the total utilization of this drug category was 1.4 times higher in the GED, and dipyrone comprised 44.2% (*n* = 46), compared with 14.4% (*n* = 15) in the OED.

#### 2.3.2. Non-Analgesic Drugs

Pain adjuvants comprised a larger proportion of administered drugs in the OED compared with the GED, (18.7% versus 10.9%, *p* < 0.001), with a similar variation in drug distribution between groups. While the OED and GED had similar usage of anti-emetics in total, drug choice in the OED was significantly more diverse (*p* < 0.001); use of metoclopramide was more predominant in the GED 50.5% (*n* = 52) than the OED 16.5% (*n* = 17), as the remaining anti-emetics in the OED were distributed between five drugs. With the exception of bisphosphonates, anticholinergics and granulocyte colony-stimulating factors, all of which target specific oncological indications, drugs in the “other” category were more commonly utilized in the GED, and these included respiratory inhalations, as well as blood pressure, diabetes, clotting and anticonvulsant agents. Of note, anticonvulsants were all used in this analysis for neurological conditions such as epileptic events, and not as pain adjuvants. Resuscitative infusions were used disparately in the two settings (*p* = 0.004), with higher usage of blood products, electrolytes and dietary supplements in the GED and more fluids in the OED. 

#### 2.3.3. Drug Administration for Pain Admissions

Figure 1 depicts drugs administration by admission type: pain and a grouping of other chief complaints. Pain was a primary complaint for 103 (17%) patients in the OED and 95 (21%) in the GED. This subgroup of patients received 48.4% (*n* = 91) and 67.0% (*n* = 69) of the total analgesics given in the GED (*n* = 188) and OED (*n* = 103), respectively. Put another way, over half of analgesics prescribed in the GED were for patients that did not report pain as a chief complaint (51.6% GED versus 33% OED). The pattern of analgesic delivery for the management of cancer pain mirrors the findings in the rest of the cohort; significantly fewer intravenous opioids were used in the OED while rapid onset opioids were only given in the OED. Moreover, according to the test of difference of proportions, with respect to pain, the use of opioids in the GED was likely to be significantly larger than OED (*p* = 0.045). Antibiotics and anticoagulants were only delivered in the GED; blood products, electrolytes and fluids were similarly administered. Pain adjuvants comprised more of the drugs delivered in the OED, including antidepressants, antipsychotics, antihistamines and steroids. 

Drug utilization by chief complaint mirrored patterns observed for pain admissions. Administration of IV opioids was significantly higher in the GED compared with the OED for patients with gastrointestinal complaints (12% vs. 2% *p* = 0.01), while new analgesic agents were used only in OED. No significant differences in pain drug usage were shown for patients presenting with emesis; however, they were more likely to receive adjuvants such as steroids in the OED. A non-significant trend for more opioids was shown in the GED across admission types, including infection, general deterioration, bleeding, neurological problems, and procedures. Finally, admissions for oncological emergencies were more likely to be treated with opioids in the OED than the GED (12.3% vs. 4.6% *p* = 0.07). Figure 2 illustrates the distribution of analgesic usage by emergency setting. 

## 3. Discussion

A fifth of all admissions to both emergency settings in this study were for the management of cancer-related pain. Universally, pain remains a dominant reason for emergency service usage among cancer patients [12], as well as a significant predictor for hospital readmission and mortality [13]. The management of cancer patients admitted to emergency services in dedicated oncology pathways has been shown to improve health care [14]. Moreover, a large population-based projection study showed that by 2060, cancer patients will die of unnecessary health related suffering unless an expansion of palliative care is integrated into cancer programs [15]. Here, subjects with matched demographics and pain-related admissions, treatment in the OED and the GED were associated with highly disparate drug delivery practices with regard to indication for treatment, route and method of administration, mechanism of action, usage of pain adjuncts and diversity in drug choice. 

In the OED, 18.5% of all delivered drugs were analgesics and 11.3% were opioids, whereas in the GED, analgesics made up 20.4% with 12.5% opioids. However, in the GED, significantly more individuals received analgesics in general and opioids specifically when admitted for reasons that did not include cancer pain. As multiple chief complaints were permitted in this study, it can be assumed that nearly all patients with cancer pain were included in this category. This may stem from a lack of knowledge or an overcrowded and fast-paced environment, but the ramifications may compromise care quality. This is sup-ported by evidence of significant differences in opioid prescribing practices across physician specialties, where emergency medicine physicians were among the most likely to prescribe opioids (OR 2.7 CI 2.6 to 2.8 compared with general practitioners) [16].

The route of drug administration significantly differed between groups. Physicians in the GED were three times more likely to prescribe intravenous opioids than those in the OED, most notably morphine. Morphine is the prototype opioid drug for moderate to severe cancer pain on the third step of the World Health Organization’s (WHO) analgesic ladder. With a large variation in patient response to morphine, studies have not demonstrated its superiority over other opioids in terms of efficacy or tolerability, and it should not be regarded as a “drug of choice” [17]. Nevertheless, its short half-life and relatively fast onset make it a convenient choice for emergency settings. Since patients were matched for pain admissions, the exorbitant usage of IV opioids in the GED suggests drug overuse and misuse. The new rapid onset analgesics with unique formulations and methods of delivery were utilized only in the OED, where oncologists possess expert knowledge of the treatment landscape. Similarly, pills, syrups and sprays were more often used in the OED.

Drug delivery in the OED can be characterized by its richer diversity when compared to the GED. The larger variety of oral opioids was presumably tailored to patient needs in the OED, while those in the GED almost always received oxycodone acetaminophen. Rotation among opioids is a useful therapeutic strategy to improve analgesic response or minimize toxicity [4]. This was also true for other classes of oncological drugs, such as anti-emetics, again suggesting the greater familiarity of oncologists with the supportive care treatment landscape as well as an effort to personalize drugs to patients. Additionally, the OED made greater usage of pain adjuvants. Essential to effective pain management, pain adjuvants enhance the potency of opioids and reduce opioid side effects [17]. Patients ad-mitted to the GED were more likely to receive drugs not recommended for the treatment of cancer-related pain. Meperidine is an opioid most often prescribed for surgical patients, with metabolites that can cause dangerous central nervous system side effects—and has no place in the cancer pain drug armamentarium. This drug was prescribed only in the GED, presumably by surgeons who make use of it in their practice with non-cancer patients. In addition, benzodiazepines were used nearly four times as often in the GED. It should be noted that the role of benzodiazepines as pain adjuvants were a long-standing controversy, in the context of potential for the development of cognitive impairment, physical and psychological dependence, worsening depression, overdose, and other side effects [18]; modern cancer pain guidelines do not include benzodiazepines [19], and an FDA safety announcement strongly warns against combining opioids with benzodiazepines because of serious risks, including death [20].

Finally, the physicians in the GED prescribed nearly double the total amount of drugs for a similar amount of patients. The patterns of drugs use and disparities in practice be-tween the OED and GED hold true both when evaluating patients with pain as a chief complaint, and also when grouping together other chief complaints. This highlights the prevalence and grave significance of the treatment of pain in cancer patients in emergency settings, both when they visit specifically due to their pain, and also when they present for other reasons. Severity and high risk admissions were more likely to be referred to the GED. These, however, did not account for disparities in pain drug usage, as no differences were observed for the high risk “neurological” complaints, while for oncological emergencies, higher opioid administration was actually observed in the OED, further suggesting a targeted treatment for accurate indications.

Not all institutions have the capacity, resources and personnel for a dedicated oncology emergency setting. Future directions for this study include the implementation of an educational intervention aimed at general emergency departments. In a prospective study, emergency care teams will undergo a pragmatic training program in supportive care treatment paradigms. Drug delivery practices as well as patient and healthcare satisfaction will be evaluated before and after the intervention. We hypothesize that improved financial and quality care parameters alongside reduced opioid usage due to the enhanced awareness and experience among GED care teams.

### 3.1. Limitations

This study has various limitations. As a single center with limited sample size, the results cannot be generalized. However, electronic medical records were reviewed individually and not automatically to ensure the accuracy of patient data and to account for multiple possible complaints. The analysis may be confounded by patient selection bias, as cases managed in the GED may be more complex that those in the OED, due to type or time of presentation (18.8% referred from OED to GED); however, the dramatic differences in drug utilization appear far greater than those accounted for by this alone. Moreover, a nationwide analysis found that patients with cancer attend emergency services more often during business hours and on weekdays and are more likely to be hospitalized during these times [21]. An additional limitation is in the lack of patient reported outcomes, or the assessment of quality measures, however hospital based surveys assessing patient satisfaction point to improved ratings after implementation of the OED. It should also be acknowledged that patients were more likely to return to the OED, and this is presumably due to their ease and comfort in admission to a more intimate care setting. 

### 3.2. Conclusions

The results of this analysis suggest that management of cancer-related pain in oncology-dedicated emergency settings is preferred to general emergency departments. Evidence based drug delivery for appropriate indications, usage of opioids, choice in method and route of drug administration, incorporation of pain adjuncts and diversity in drug choice were all enhanced in the oncology-based setting. Efforts should be directed toward the reduction of disparities in urgent cancer care, and can be achieved by increasing emergency physician knowledge of supportive care treatment paradigms, expanding outpatient symptom management programs, and, finally, the integration of urgent care services in cancer programs. As such, optimized urgent cancer care may contribute to superior patient and healthcare outcomes.

## 4. Materials and Methods

### 4.1. Setting

A large tertiary referral center for 12 district hospitals, RMC serves approximately 20% of the total population of Israel, and patients with cancer have direct access to inpatient, outpatient, and emergency services covered by national healthcare. Patients presenting during clinic hours visit the OED pending availability of staff, resources and beds, without initial selection or triage; when the OED is filled or closed, they present directly to the main hospital GED, where an oncology consult can be obtained if needed. Open during weekday morning hours (9:00 a.m. until last bed or 12:00 p.m.), the OED is staffed with a designated nurse and oncologist. A palliative care nurse is usually available for consultation. Resources shared with the day care include drugs and fluids for patient administration, while imaging is shared with the GED. After treatment, patients are discharged, hospitalized in the oncology inpatient department, or referred to the GED for more advanced multi-disciplinary evaluation and care.

### 4.2. Subjects

A retrospective electronic medical record review was performed for all patients admitted to the GED and OED during the 3-month period between April 2017 and June 2017. Data for the GED were obtained only for days the OED was open in order to establish similar groups and exclude confounders in weekend or holiday visits. Patients could attend the GED at any hour and needed to have a documented oncology consult. Hematologic malignancies were excluded since they are treated in a separate department at RMC. Patient demographics and disposition (hospitalization, discharge, referral) were retrieved from the database. Drug administration was also recorded and divided into specific drug group categories. Cases were assigned “chief complaint” after individual file evaluation by the study team, and could be given more than one assignation, for instance “oncological emergency” and “neurological symptoms”. Patients referred from the OED to the GED were considered in both settings as separate cases but this was taken into account for statistical analysis. The study was approved by the institutional Independent Ethics Committee, registered as 0465-17-RMB and granted a waiver for obtaining patient consent because of its retrospective design.

### 4.3. Statistical Analysis

To assess the difference between drug administration over the emergency setting of interest we implemented a series of chi-square tests or Fisher’s exact tests (when the assumptions of the parametric chi-square test were not met) and nonparametric Mann–Whitney U tests. The results were statistically significant by the standard of the study when *p* was <0.05.

## Figures and Tables

**Figure 1 pharmaceuticals-15-00805-f001:**
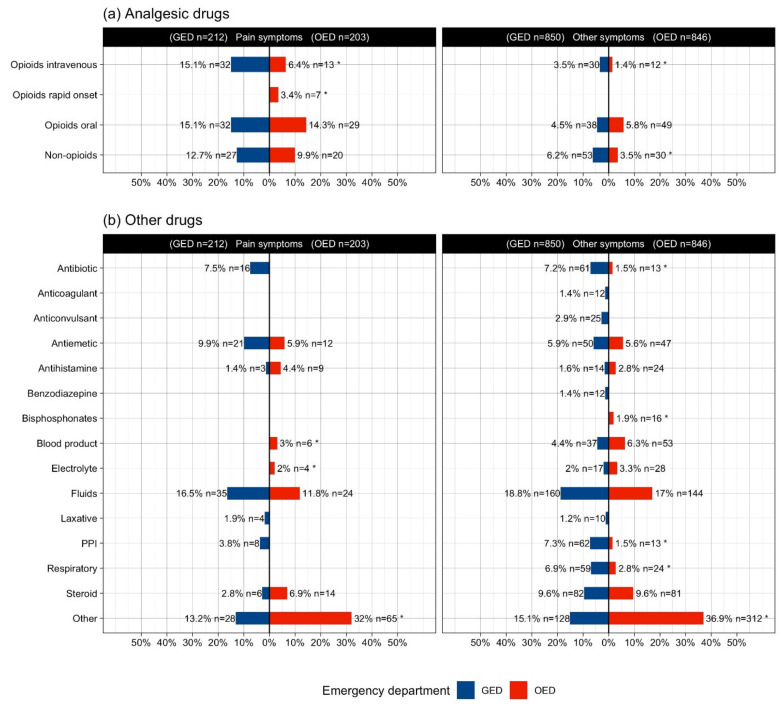
Title: Drug administration for pain and other symptoms according to emergency setting. Legend: Practices of drug administration significantly differed by emergency setting, both for pain and other symptoms. Section (**a**) refers to direct pain drugs, while (**b**) to non-pain medications. GED—General emergency department; OED—Oncology emergency department. * denotes statistical significance.

**Figure 2 pharmaceuticals-15-00805-f002:**
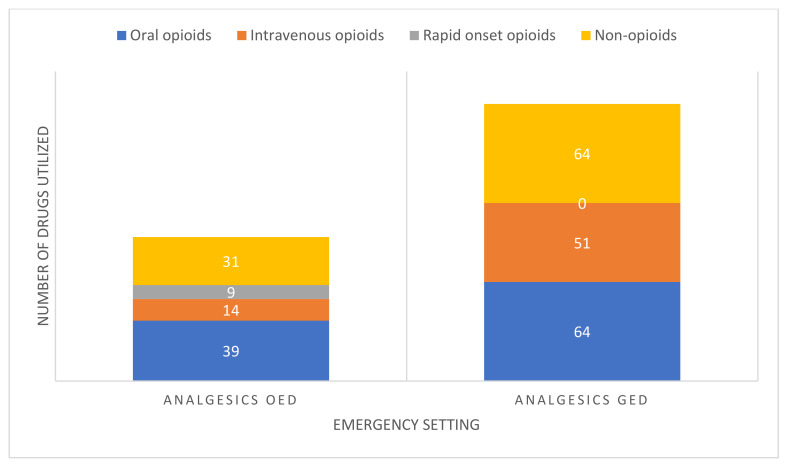
Title: Analgesic utilization in GED and OED. Legend: Among the analgesics, the use of oral opioids, intravenous opioids and rapid onset opioids in OED was significantly different from GED (*p* < 0.001, *p* = 0.003, *p* = 0.003, respectively). GED—General emergency department; OED—Oncology emergency department.

**Table 1 pharmaceuticals-15-00805-t001:** General demographics of the patients visiting the emergency settings from April to June 2017.

Demographics	Level	GED	OED	*p*
*n*		443	479	
Gender (%)	Male	238 (53.7)	271 (56.6)	
City (%)	Hospital city (Haifa)	125 (28.7)	191 (40.0)	<0.001
Age (mean (SD))		63.66 (17–92)	62.78 (23–89)	0.32
Hours in ED (mean (range))		10.08 (6.88)		
Disposition (%)	Discharge	198 (44.7)	343 (71.6)	<0.001
	Hospitalize	245 (55.3)	46 (9.6)	
	Refer	0 (0.0)	90 (18.8)	
Diagnosis (%)	Breast		83 (17.4)	
	Central nervous system, head and neck, gynecology, other		30 (6.3)	
	Gastrointestinal		128 (26.8)	
	Genitourinary		93 (19.5)	
	Lung and other thoracic		114 (23.8)	
	Melanoma and skin		20 (4.2)	
	Sarcoma		10 (2.1)	
Admissions per patient (%)	1	221 (49.9)	165 (34.4)	<0.001
	2	118 (26.6)	132 (27.6)	
	3–6	90 (20.3)	148 (30.9)	
	6>	14 (3.2)	34 (7.1)	
Chief complaint (%)	Total complaints (multiple per admission) = *n*	452	605	
	Oncology emergency = yes (%)	54 (11.9)	30 (5.0)	<0.001
	Neurological = yes (%)	55 (12.2)	49 (8.1)	0.036
	Infection = yes (%)	34 (7.5)	46 (7.6)	1
	Other = yes (%)	30 (6.6)	55 (9.1)	0.181
	Pain = yes (%)	95 (21.0)	103 (17.0)	0.117
	General deterioration = yes (%)	48 (10.6)	52 (8.6)	0.314
	Respiratory = yes (%)	49 (10.8)	50 (8.3)	0.188
	Bleed and thromboembolic = yes (%)	19 (4.2)	20 (3.3)	0.548
	Gastrointestinal = yes (%)	36 (8.0)	52 (8.6)	0.799
	Procedure = yes (%)	13 (2.9)	125 (20.7)	<0.001
	Emesis = yes (%)	19 (4.2)	22 (3.6)	0.755

GED—General emergency department; OED—Oncology emergency department.

**Table 2 pharmaceuticals-15-00805-t002:** Administration of drugs in the emergency settings: general emergency department (GED) vs. oncology emergency department (OED).

Drug Category	Drug Sub-Category	Drug	GED (*n*)	OED (*n*)	GED (%)	OED (%)	*n*	Statistic	*p*	df	p.Signif
All drugs			921	557	62.3	37.7	1478				
Analgesics	Oral opioids	Morphine immediate release	0	10	0.0	8.8	113	73.29	0.0000	5	****
Analgesics	Oral opioids	Oxycodone liquid	1	25	0.9	22.1					
Analgesics	Oral opioids	Oxycontin	3	0	2.7	0.0					
Analgesics	Oral opioids	Oxycodone acetaminophen	58	9	51.3	8.0					
Analgesics	Oral opioids	Oxycodone naloxone	0	3	0.0	2.7					
Analgesics	Oral opioids	Tramadol flashtabs	2	2	1.8	1.8					
Analgesics	Intravenous opioids	Meperidine	10	0	15.4	0.0	65	13.68	0.0034	3	**
Analgesics	Intravenous opioids	Fentanyl	0	1	0.0	1.5					
Analgesics	Intravenous opioids	Morphine	41	11	63.1	16.9					
Analgesics	Intravenous opioids	Tramadol	0	2	0.0	3.1					
Analgesics	Nonopioids	Diclofenac sodium	5	3	4.8	2.9	104	7.76	0.2560	6	ns
Analgesics	Nonopioids	Dipyrone	46	15	44.2	14.4					
Analgesics	Nonopioids	Ibuprofen	1	0	1.0	0.0					
Analgesics	Nonopioids	Papaverine	14	6	13.5	5.8					
Analgesics	Nonopioids	Paracetamol	6	3	5.8	2.9					
Analgesics	Nonopioids	Mebeverine	0	1	0.0	1.0					
Analgesics	Nonopioids	Naproxen Sodium	1	3	1.0	2.9					
Analgesics	Rapid onset opioids		0	9	0.0	100.0	9	9.00	0.0027	1	**
Gastrointestinal	Antiemetic	Netupitant/palonosetron	0	9	0.0	8.7	103	31.27	0.0000	5	****
Gastrointestinal	Antiemetic	Aprepitant	0	4	0.0	3.9					
Gastrointestinal	Antiemetic	Sulpiride	0	1	0.0	1.0					
Gastrointestinal	Antiemetic	Palonosetron	0	1	0.0	1.0					
Gastrointestinal	Antiemetic	Metoclopramide	52	17	50.5	16.5					
Gastrointestinal	Antiemetic	Ondansetron	8	11	7.8	10.7					
Pain adjuvants	Anesthesia		1	1	0.5	0.5	204	6.74	0.1500	4	ns
Pain adjuvants	Antidepressant		0	3	0.0	1.5					
Pain adjuvants	Antihistamine		16	27	7.8	13.2					
Pain adjuvants	Antipsychotic		2	3	1.0	1.5					
Pain adjuvants	Steroid		81	70	39.7	34.3					
Resuscitative infusions	Blood product		30	50	7.0	11.7	426	13.51	0.0037	3	**
Resuscitative infusions	Dietary supplements		1	3	0.2	0.7					
Resuscitative infusions	Fluids		174	125	40.8	29.3					
Resuscitative infusions	Electrolyte		19	24	23.8	5.6					
Anti-infectives	Antibiotic		65	13	15.3	3.1	80	4.26	0.0390	1	*
Anti-infectives	Antifungal		0	2	0.0	0.5					
Gastrointestinal	Antidiarrheal		0	5	0.0	1.2	21	4.73	0.0297	1	*
Gastrointestinal	Laxative		11	5	3.1	1.2					
Other	Antacid		3	0	0.7	0.0	353	66.84	0.0000	10	****
Other	Anticholinergic		0	2	0.0	0.5					
Other	Anticoagulant		14	3	3.3	0.7					
Other	Anticonvulsant		26	0	6.1	0.0					
Other	Benzodiazepine		15	4	3.5	0.9					
Other	Bisphosphonates		1	15	0.2	3.5					
Other	Granulocyte colony stimulating factor		2	2	0.5	0.5					
Other	Antihypertensive and diabetes		48	12	11.3	2.8					
Other	Proton pump inhibitor		66	12	15.5	2.8					
Other	Respiratory inhalation		50	18	11.7	4.2					
Other	Other		48	12	11.3	2.8					

*, **, **** denotes statistical significance.

## Data Availability

Data is contained within the article.

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
