# Peer review of "Approach to Cancer Pain Management in Emergency Departments: Comparison of General and Oncology Based Settings"

_pharmaceuticals, 2022, doi:10.3390/ph15070805_

Round 1

Reviewer 1 Report

The article “One goal-different ways: treatment of cancer pain in general and oncology emergency departments” is very interesting and very well written. In the Material and Method section, the authors state that “Acute pain associated with oncological emergencies was not included.” Could you please add some details as to why?

Minor comments

Introduction – “In the midst of an opioid crisis, new restrictive regulations have generally exempt patients with cancer, however for some, limited access has been an unintended consequence [6].” – slightly unclear, please rephrase

Author Response

Dear Reviewer 1,

Thank you for your review of our manuscript, your approval and helpful comments. 

1. Thank you for drawing our attention to this sentence which is unclear and misleading to the reader. We have decided to remove it. The original point was that patients with oncological emergencies were assigned the additional complaint “pain” only if it was a presenting symptom, and not assumed -- however this is redundant. 

2. Rephrased sentence, hopefully now more clear: “Restrictive regulations on opioid usage have generally exempt oncological settings, however some patients with cancer experience limited access as an unintended consequence of the ongoing opioid crisis [6].” 

Sincerely ,

Ilit Turgeman

Reviewer 2 Report

The authors presented data of significance highlighting the importance of personalized pain management in cancer patients. The work demonstrates a disparity in OED and GED pain care.  

The authors in their previous report suggested a “GED recommendation for high-risk patients for a better advanced multidisciplinary management”. Current work, a continuation of conclusions from previous work though stresses on significance of OED referral doesn’t shed light on the above statement on GED from their last publication which may be a possible explanation for excessive dose prescription unless studied. It would be interesting to classify the data based on severity/risk and may provide clarity on this disparity and help document the observations on high-risk patients.

Including the future directions of the study in the discussion may help attract readers attention and provide valuable insights.

Author Response

Dear Reviewer 2,

We greatly appreciate your review and helpful comments.

  1. Thank you for bringing up this point of classifying patients due to severity /risk especially in light of our previous conclusion regarding referral of specific patients to the GED, perhaps explaining the disparities in drug usage. We actually originally constructed Figure 1 for each chief complaint, however the results were extremely busy and we didn’t find that they contributed significantly to the work.  Your comment reinforces the importance in finding a way to explain these differences, so we have now reevaluated the original figures and made an effort to put them into writing in the following paragraph, added to the "Results" section:   Drug administration by admission type   Drug utilization by chief complaint mirrored patterns observed for pain admissions. Administration of IV opioids was significantly higher in the GED compared with the OED for patients with gastrointestinal complaints (12% vs 2% p=0.01), while new analgesic agents were used only in OED. No significant differences in pain drug usage were shown for patients presenting with emesis, however, they were more likely to receive adjuvants such as steroids in the OED. A non-significant trend for more opioids was shown in the GED across admission types, including infection, general deterioration, bleeding, neurological problems, and procedures. Finally, admissions for oncological emergencies were more likely to be treated with opioids in the OED than the GED (12.3% vs 4.6% p =0.07).   The following has been added to the "Discussion":   Severity and high risk admissions were more likely to be referred to the the GED. These, however, did not account for disparities in pain drug usage, as no differences were observed for the high risk “neurological" complaints, while for oncological emergencies, higher opioid administration was actually observed in the OED, further suggesting targeted treatment for accurate indications.   
  2. Future directions: Not all institutions have the capacity, resources and personnel for an oncology dedicated emergency setting. Future directions for this study include the implementation of an educational intervention aimed at general emergency departments. In a prospective study, emergency care teams will undergo a pragmatic training program in supportive care treatment paradigms. Drug delivery practices as well as patient and healthcare satisfaction will be evaluated before and after the intervention. We hypothesize improved financial and quality care parameters alongside reduced opioid usage due to the enhanced awareness and experience among GED care teams.  

Sincerely,

Ilit Turgeman